# Construction and Application of Materials Knowledge Graph in Multidisciplinary Materials Science via Large Language Model

Yanpeng Ye[1,2,†], Jie Ren[2,3,†], Shaozhou Wang[2,*], Yuwei Wan[2,4], Imran Razzak[1], Bram Hoex[5], Haofeng Wang[6], Tong Xie[2,5,*], and Wenjie Zhang[1,*]

[1]School of Computer Science and Engineering, University of New South Wales
[2]GreenDynamics
[3]Department of Materials Science and Engineering, City University of Hong Kong
[4]Department of Linguistics and Translation, City University of Hong Kong
[5]School of Photovoltaic and Renewable Energy Engineering, University of New South Wales
[6]College of Design & Innovation, Tongji University

## Abstract

Knowledge in materials science is widely dispersed across extensive scientific literature, posing significant challenges to the efficient discovery and integration of new materials. Traditional methods, often reliant on costly and time-consuming experimental approaches, further complicate rapid innovation. Addressing these challenges, the integration of artificial intelligence with materials science has opened avenues for accelerating the discovery process, though it also demands precise annotation, data extraction, and traceability of information. To tackle these issues, this article introduces the Materials Knowledge Graph (MKG), which utilizes advanced natural language processing techniques integrated with large language models to extract and systematically organize a decade's worth of high-quality research into structured triples, contains 162,605 nodes and 731,772 edges. MKG categorizes information into comprehensive labels such as Name, Formula, and Application, structured around a meticulously designed ontology, thus enhancing data usability and integration. By implementing network-based algorithms, MKG not only facilitates efficient link prediction but also significantly reduces reliance on traditional experimental methods. This structured approach not only streamlines materials research but also lays the groundwork for more sophisticated science knowledge graphs.

## 1 Introduction

In the contemporary information era, despite notable advancements, the creation and advancement of novel materials still heavily rely on traditional, time-consuming trial-and-error methods intertwined with chemical and physical intuitions. These conventional research approaches significantly impede the life-cycle of high-performance material research. Given the specialization, inherent complexity, and vast knowledge base of material science, researchers focusing on a single direction often struggle to efficiently access and understand material knowledge from multidisciplinary studies. For instance, researchers in solar cell development might not fully comprehend studies related to solid-state

---

*Corresponding authors: shaozhou@greendynamics.com.au; tong@greendynamics.com.au; wenjie.zhang@unsw.edu.au

†These authors contributed equally to this work.

batteries or organic light-emitting diodes. Yet, the electronic properties of materials across these different domains are highly related, and researchers in different domains can potentially learn from each other. To accelerate the progress of materials research, there is a pressing need to efficiently integrate knowledge from various disciplines [1]. However, this vital knowledge is scattered across a vast array of over 10 million scientific papers, covering diverse topics and disciplines such as materials preparation and functionalization methods, advanced materials characterization techniques, and the exploration of physical, chemical, and biological properties, along with their applications in fields like electronic devices, clean energy storage and transfer, and mechanical engineering. This fragmentation of knowledge represents a significant barrier to interdisciplinary collaboration and innovation. A critical gap in current research infrastructure is the lack of an effective materials science database that can consolidate this scattered knowledge, facilitating easier access and interdisciplinary integration.

Despite the existence of current databases of scientific literature such as Scopus, Web of Science, and Crossref, which offer ways to search for research papers based on specific labels, extracting useful information about material science from the vast ocean of literature remains demanding. To obtain a clearer sense of materials properties, some structured database projects such as Materials Project [2], OQMD [3], and NOMAD [4] were developed. However, these databases contain many computational results obtained through techniques like Density Functional Theory (DFT) or Molecular Dynamics (MD) simulations [5]. While these computational databases can provide valuable references for predicting and understanding certain materials systems, they often face discrepancies with experimental observations. Therefore, there is an urgent need within the field of materials science for a database grounded in experimental research and practical information.

Knowledge graph (KG) is a structured representation of information that models the controlled vocabulary and ontological relations of a topical domain as nodes and edges, enabling complex queries and insights that traditional databases cannot easily provide. The adoption of knowledge graphs offers several advantages, including enhanced data interoperability, the ability to infer new knowledge through relational data analysis, and improved data quality and consistency through structured representation [6], [7]. These features make knowledge graphs particularly valuable for integrating diverse information sources and providing a unified view of a domain's knowledge, thereby facilitating more informed decision-making and discovery [8]. However, the construction of knowledge graphs in specific fields always requires the participation of a large number of experts [9]. This labor-intensive process not only limits the scalability of KGs but also impacts their performance and timeliness [10]. With the rapid development of natural language processing (NLP), methods for extracting information from unstructured text and constructing knowledge graphs have become more efficient and accurate [11]. For instance, in 2016, the Metallic Materials Knowledge Graph (MMKG) was developed to store materials information from various web data resources [12]. Knowledge graphs tailored to lithium-ion battery cathodes have been constructed, aimed at identifying potential new materials candidates [13]. User-friendly databases focusing on specific material types, such as Metal-Organic Framework Knowledge Graphs (MOF-KG), have been developed [14]. Recently, a material knowledge graph, MatKG, and MatKG2, containing information on material properties, structure, and applications, has been developed [1], [15].

However, these material knowledge graphs face even greater challenges. Firstly, although advancements in NLP technology have reduced the dependency on experts to a certain extent, training data still requires extensive annotation to enhance model accuracy [16]. Secondly, the construction of these knowledge graphs often involves predicting relationships between nodes to form triples, which means the entities represented in the KG are not always based on real instances [17]. This can diminish the authenticity and credibility of the KG. Additionally, this approach makes updating the knowledge graph difficult, as each new node introduced necessitates predicting its relationship with every other node, complicating the maintenance of a dynamic and accurate knowledge graph, especially in advanced fields like material science. Acknowledging these challenges, the emergence of LLMs like GPT and LLaMA represents a breakthrough, offering new solutions to enhance the zero-shot method [18], extraction, and credibility of structured information [19], [20]. The fine-tuning technique of LLMs can significantly enhance their performance in specific domain text tasks through training with fewer samples [21],[22]. This means improving the results of Named Entity Recognition (NER) and Relation Extraction (RE) without requiring a large amount of labor becomes possible and was adopted in our research.

In this paper, we have achieved significant advancements in the development of Materials Knowledge Graph (MKG), a pioneering graph database tailored for the field of materials science. Our contributions are highlighted in three key areas: 1) We propose a method to achieve NER, RE, and entity resolution (ER) with high accuracy. Through this method, we can easily convert unstructured text into triples and retain the source information of each triplet. This method also makes updating KG very convenient. 2) We constructed the first accurate knowledge graph dedicated to materials, where researchers can easily get information about the material by querying the MKG. 3) We use a well-defined label system so our KG can be easily scaled up and potentially combined with other structured databases or KGs. 4) We demonstrate a similarity calculation method based on Jaccard Similarity for materials and applications.

## 2 Methods

Figure 1 presents the elaborated pipeline of our study. The KG construction part can be divided into there tasks - ontology design, knowledge extraction, and entity resolution (ER). For domain-specific KG, the design of ontology often relies on experts in the field. This work forms an effective ontology by defining and summarizing a small number of papers through LLM. The knowledge extraction task begins with the manual annotation and normalization of the initial training dataset, annotated training set is used to finetune the LLM for NERRE tasks. Simultaneously, the inference dataset is divided into ten batches, facilitating the iterative refinement process that follows. The ER tasks are conducted using advanced NLP technologies, including ChemDataExtractor [23], mat2vec [24], and our expertly curated dictionary. These steps enable the integration of information from these distinct fields into a unified knowledge graph and clean the extracted data. After ER, we selectively enhance the training dataset with high-quality results to improve the model's performance in subsequent iterations. The knowledge graph is finally constructed using the normalized data from the last iteration. To complete the graph and predict potential material applications, we employ both network-based algorithms and graph embeddings. This methodology provides critical insights and recommendations for researchers in the materials science domain.

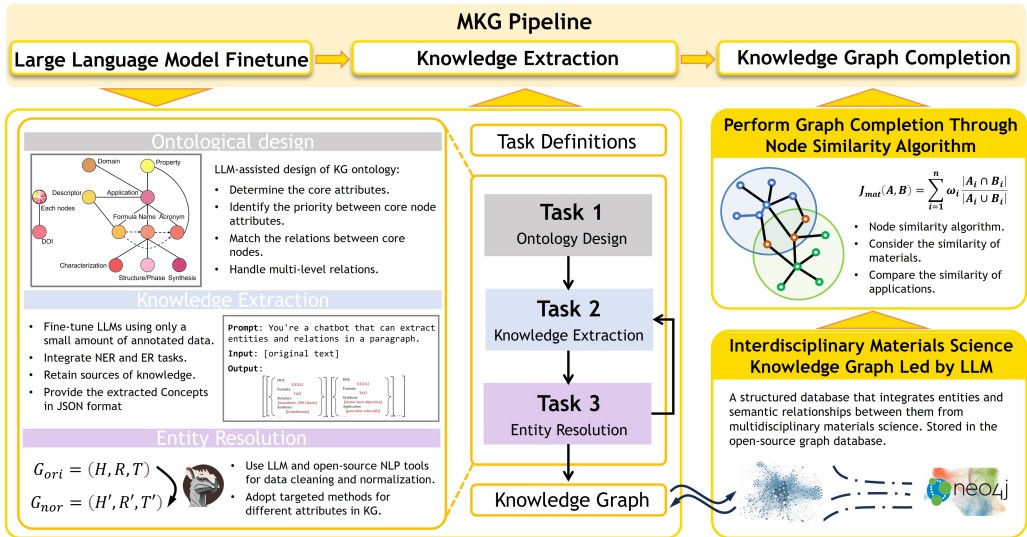

Figure 1: Pipeline of the fine-tuned LLM for knowledge graph tasks.

### 2.1 Data preparation and schema design

Material experts annotated nine distinct categories from the abstracts of 75 research papers, forming the training dataset for the Large Language Models (LLMs). The structure of the data reflects the structure of the KG. This structure summarizes ten articles through LLM and extracts key attributes. As illustrated in Figure 2 (a), the "Formula", "Name" and "Acronym" node serves as the central hub within the graph, linking to nodes that encapsulate its nomenclature, composition, and various attributes. Among these core attributes, the priority of the attribute pointed to by the

arrow decreases in sequence. The "Structure/Phase" node describes the material's physical state, while the "Application" node denotes its practical uses, and the "Property" node details its inherent characteristics. Additionally, the "Descriptor" node provides qualitative information enhancing the contextual understanding of the material.

Furthermore, the "Application" node is extended to include "Property", "Descriptor", and "Domain" nodes, indicating the specific attributes and the broader context of the material's application. Among them, "Domain" is unique to "Application" and represents the field to which the application belongs. To maintain data provenance and traceability, each node is linked to a "Digital Object Identifier (DOI)" node. This allows for source verification where querying the intersections of "DOI" node connections at both ends of a relation can pinpoint the source article from which the relations were derived, ensuring data integrity within the graph structure. Figure 2 (b) is an example of MKG, from which it can be seen that the core of MKG is to capture the connection and structure between materials and their applications.

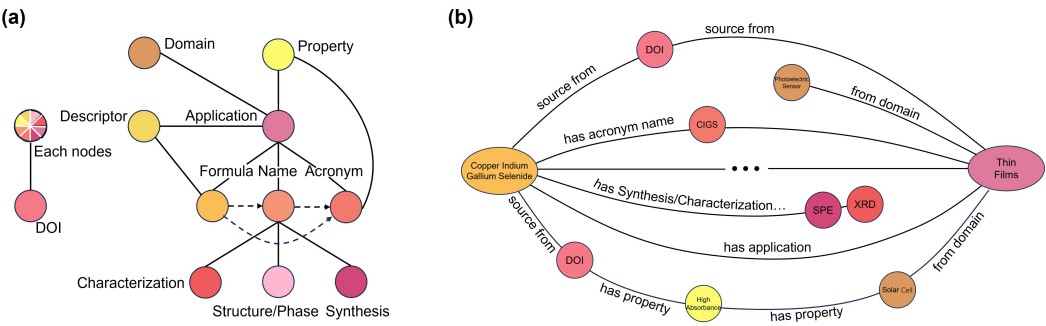

Figure 2: This schematic represents the (a) MKG schema and (b) an example of path in MKG between the "Name" node "Copper Indium Gallium Selenide" and "Application" node "Thin Films".

Given the inherent complexity of sentences and the variability in terminologies across abstracts, a normalization process was employed after the initial extraction. This normalization ensured a uniform representation of entities with similar meanings. For example, terms such as "Lithium-Ion Battery" and "Li-ion batteries" were standardized to "lithium-ion battery", while phrases like "solution casting method", "solvent post-treatment method", and "solution-based deposition" were simplified to "solution-processed". All annotated entities underwent this normalization process, ensuring consistency and facilitating effective training of LLMs. The field of materials encompasses a wide range of areas. At this stage, our priority is to focus on energy materials. We downloaded 150,000 abstracts of peer-reviewed research articles on energy material science, including batteries, solar cells, and catalysts, from the Web of Science. Each abstract was stored in a JSON file format, structured as "DOI - text", facilitating seamless processing and analysis.

## 2.2 LLMs training, evaluation and inference

The training dataset, composed of compiled data, was employed to fine-tune models including LLaMA 7b, LLaMA2 7b [25], and Darwin [26]. Upon obtaining high-quality results from the normalized inference, we iteratively retrain a fine-tuned LLM to infer subsequent batches of data. The models underwent training over 10 epochs with a batch size of 1. Additionally, 60 abstracts were annotated to assess the LLMs" performance. Our evaluation primarily focused on the NER capabilities of the model and preliminarily explored the RE task's ability to identify potential internal relations among relevant element sets. Moreover, since we standardized entities during the data compilation phase, we also assessed the LLMs" effectiveness in standardizing the entities. Specifically, for NER, RE, and ER tasks, we employ a unified framework for evaluation based on precision, recall, and the F1 score, taking into account the instances of false positives (FP) and false negatives (fn) to quantify the performance.

For NER, a true positive is a correctly identified entity, while for RE, it is a correctly identified relationship between entities, and for ER, it is a correctly standardized entity according to the schema. Conversely, a false positive occurs when the model incorrectly identifies an entity, relation, or standardization, and a false negative is when the model fails to identify a correct entity, relation, or

schema element that should have been recognized. After defining these terms, we can evaluate each task using standard precision, recall, and F1 score metrics.

After evaluation, we chose a fine-tuned LLM that demonstrates optimal performance in both NER and RE tasks to iteratively infer the 150,000 abstracts. The output of inferences is organized in the "DOI—text—response" format. Consequently, the fine-tuned LLM not only extracts entities but also assigns them appropriate labels, thereby accomplishing NER and RE tasks concurrently. Moreover, every entity and relation identified in the response is traceable, enhancing the integrity and utility of the data.

## 2.3 Entity resolution

The quality of KG is crucial for its credibility in checking and correcting the inference results before graph construction. To ensure the precision of these results, we initially employed ChemDataExtractor to identify chemical formulas and "Name-Acronym" pairs from abstracts. Subsequently, entities recognized by both the Large Language Model (LLM) and ChemDataExtractor are embedded using the mat2vec model. We analyze their similarities to rectify core entities and ensure accurate "Name" to "Acronym" associations. Through this step, we can make the "Name" and "Formula" different from "Acronym" in MKG. Therefore, we have named this stage "*ER-NF/A*" ("Entity resolution - Name/Formula and Acronym"). Given the frequent mislabeling of "Name" and "Formula", we progress to the "Entity resolution - Name and Formulas" ("*ER-N/F*") phase. Here, we refine the LLM using a specifically curated training set comprising 2,000 accurately labeled entities for binary classification, which is evaluated against an additional set of 200 labels.

For other labels, we implement a Density-Based Spatial Clustering of Applications with Noise [27] algorithm to create the "Entity resolution - expert dictionary" ("*ER-ED*"). This algorithm dynamically forms clusters based on vector similarity without the need to predefine the number of clusters. Each cluster is named by material science experts, leading to the creation of an expert dictionary containing approximately 600 terms related to structures, phases, applications, and more. This dictionary, along with the entities extracted by the LLM, undergoes similarity analysis to standardize and confirm the accuracy of both entities and relations. To elevate the quality of the training dataset, we selectively integrate high-quality data from normalized inference outputs from each iteration into the training set, continuously monitoring and enhancing the performance of the fine-tuned LLM to ensure its efficacy.

## 2.4 Knowledge graph construction

Given a collection of entities $E$ and relations $R$, a knowledge graph $\mathcal{K} = E \times R \times E$ is structured as a directed multi-relational graph. It comprises triples formatted as $(h, r, t) \in \mathcal{K}$, where $h$ and $t$ are entities within $E$. To structure the inference results into triples, we employ three labels that signify the material as the core label, with the core label serving as the head $h$, the names of other labels forming the relations $R$, and their values acting as tails $t$. Within the hierarchy of core labels, "Formula" is accorded the highest priority, followed by "Name", and then "Acronym". Furthermore, each head $h$ and tail $t$ node is linked to the DOI of the source article associated with the triplet. This setup allows us to ascertain the provenance of the relation between any two nodes by examining the intersection in their connected entities. Then we transfer the triples into MKG and store the MKG via graph database Neo4j, which also supports the subgraph matching function, where subgraph matching naturally suits the need to search for certain materials with user-input conditions. To facilitate access to the detailed information, we also make the dataset available in the RDF and CSV format for straightforward data handling.

## 2.5 Graph completion

The process of Graph Completion (GC) is shown in Figure 3 (a), where we perform link prediction through GC, segmented into four primary stages: graph splitting, similarity calculation, validation and evaluation, and parameter optimization. This structured approach ensures a comprehensive exploration of the link prediction capabilities. Graph splitting is meticulously performed based on the chronological assignment of the nodes. Nodes encapsulated within a defined prediction window $\delta$ are utilized as $G_{\text{ver}}$, which are designated for the validation of predictions. Nodes preceding this predictive interval serve as $G_{\text{tra}}$, employed to train and refine the prediction models. During the similarity calculation stage, advanced graph algorithms and embeddings are applied on $G_{\text{tra}}$ to

ascertain the similarity between the "Materials" and "Applications" nodes. This phase leverages an enhanced Jaccard similarity metric, partitioned into three distinct components: $S(m,a)$, $F(m,a)$, and $T(m,a)$, which collectively improve the specificity of predictions.

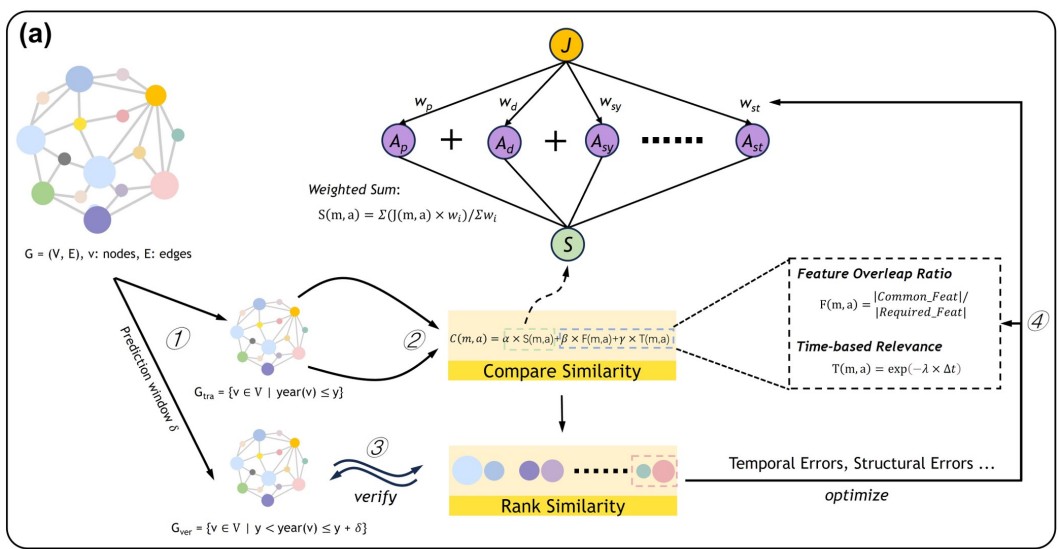

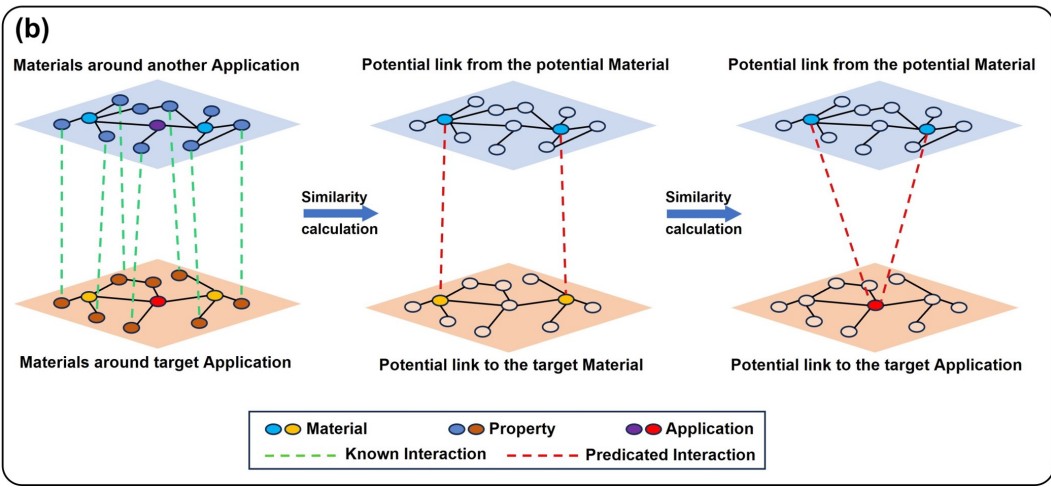

Figure 3: (a)The process of MKG graph completion and (b) the schematic diagram of nodes comparison.

In the validation and evaluation phase, the sorted "Material" and "Application" nodes undergo rigorous testing on $G_{ver}$. The effectiveness of the predictive model is assessed by tallying the incorrect predictions, represented as En, which serves as a crucial measure of performance. The final stage, parameter optimization, involves the meticulous adjustment of the parameters $S(m,a)$, $F(m,a)$, and $T(m,a)$ based on their performance on $G_{ver}$. This iterative process is crucial for minimizing the count of En and enhancing the overall accuracy of the GC method. This adaptive adjustment underscores the dynamic and responsive nature of the model in refining link prediction accuracy within the graph completion framework.

It is worth mentioning that from a global perspective, we not only need to consider whether an "Application" may be similar to a "Material" node. We also need to consider whether the "Material" currently associated with this "Application" is similar to the potential "Material". Therefore, further improvement of Jaccard Similarity:

$$J_{\text{mat}}(A, B) = \sum_{i=1}^{n} w_i \cdot \frac{|A_i \cap B_i|}{|A_i \cup B_i|}$$

where $n$ is the number of attributes considered, $A_i$ and $B_i$ are the specific attributes of materials $A$ and $B$ respectively, and $w_i$ are the weights assigned to each attribute, signifying their importance in the context of material application. As shown in Figure 3 (b), this approach mimics the investigative processes commonly employed by materials scientists when developing new materials. In determining the suitability of new materials for specific applications, researchers systematically analyze their chemical and physical properties, as well as their structural characteristics, comparing these attributes to those of materials well-established in the target field. This enhanced method not only accounts for the number of overlapping attributes but also emphasizes the significance of each attribute in relation to the material's potential application. A higher $J_{\text{modified}}$ score indicates a stronger likelihood of applicability between the material in question and the target application. Thus, materials scoring high on this metric are considered promising candidates for the specified applications.

In addition to the Enhanced Jaccard Similarity, we employ the TransE model [28]. TransE treats entities and relations as vectors in the same embedding space. The core idea of TransE is to model relations by interpreting them as translations in the embedding space.

# 3 Result

To illustrate the advantages of MKG more clearly, we conducted a comparative analysis with MatKG2 [15], highlighting the differences in their construction methodologies as illustrated in Figure 4. MatKG2 is built using a multi-step process for named entity recognition (NER) and relation extraction (RE) that does not retain the origin of each relation. In contrast, our approach employs an end-to-finish methodology utilizing a single large language model (LLM) designed to handle both NER and RE simultaneously. This not only preserves the source of each triple within the knowledge graph, enhancing its factual accuracy, but also demonstrates superior performance in the RE task.

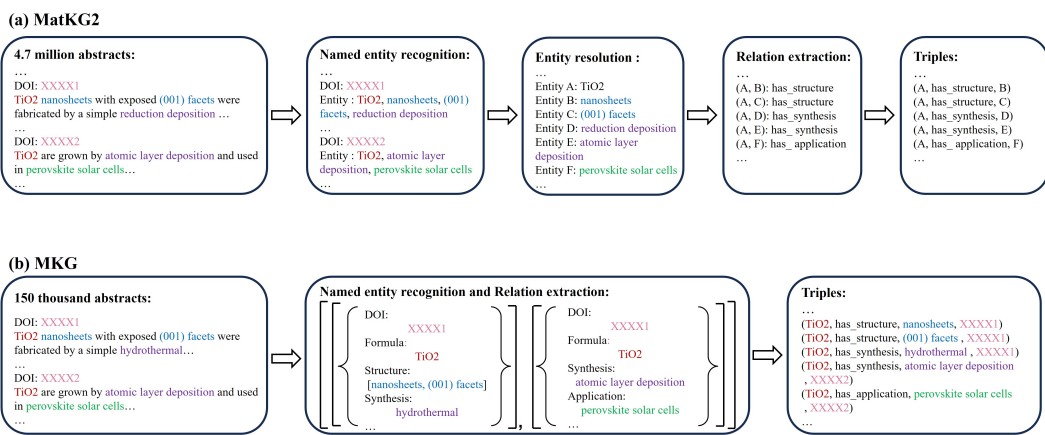

Figure 4: Schematic comparison of MKG and MatKG2.

To demonstrate the effectiveness of this pipeline, we evaluated the performance of each LLM on each task, as shown in Table 1. Darwin significantly outperforms both the LLaMA 7b and LLaMA2 7b models in the F1 scores for NER and RE tasks, suggesting that Darwin yields more effective results in text-related tasks in materials science. However, there is no marked difference in the performance of these models on ER tasks; we can say LLM has a weaker ability to complete ER in our task. This may be attributed to the limited contextual memory capabilities of the LLMs, and this is why we need to apply additional ER processes. The normailzed Darwin shows the performance of our ER and normalization process, the result indicates that it not only achieves the ER task successfully, but also contributes to the NER and RE task.

To better understand the enhancements provided by each component in the ER/Normalization process, ablation studies were performed. The outcomes, detailed in Table 2, demonstrate that each technique

Table 1: Result of NER, RE, and ER through Fine-tuned LLMs.

| Model | Task | Precision | Recall | F1 score |
|---|---|---|---|---|
| **MatBERT** | NER | 0.1196 | 0.6869 | 0.2036 |
| | RE | 0.0250 | 0.5696 | 0.0479 |
| | ER | 0.0928 | 0.5303 | 0.1579 |
| **Llama 7b** | NER | 0.6101 | 0.6216 | 0.6158 |
| | RE | 0.5305 | 0.5405 | 0.5355 |
| | ER | 0.3687 | 0.3757 | 0.3722 |
| **Llama2 7b** | NER | 0.7419 | 0.7667 | 0.7541 |
| | RE | 0.6452 | 0.6667 | 0.6557 |
| | ER | 0.4484 | 0.4633 | 0.4557 |
| **Darwin** | NER | 0.8013 | 0.7935 | 0.7974 |
| | RE | 0.7036 | 0.6968 | 0.7002 |
| | ER | 0.4593 | 0.4548 | 0.4571 |
| **Darwin (ER)** | NER | 0.9520 | 0.9083 | 0.9296 |
| | RE | 0.9039 | 0.8625 | 0.8827 |
| | ER | 0.9127 | 0.8708 | 0.8913 |

positively affects the effectiveness of ER. The results highlight that utilizing the expert dictionary (*ER-ED*) is the most beneficial method, enhancing accuracy comprehensively across nearly all labels that are included in the dictionary. Additionally, the improvements seen with *ER-NF/A* are noteworthy; this step is based on the ChemDataExtractor, effectively removes the majority of incorrect material identifications in core labels and offers even more substantial benefits to NER than to *ER-ED*. Furthermore, from the results of the ablation experiment, *ER-N/F* appears that its contribution is not as significant as the other two components, but considering the confusion between the material Formula and Name may have an impact on subsequent MKG applications. *ER-N/F* is also an indispensable part of the *ER* process. The normalized data is transformed into triples to construct the MKG, which comprises 162,605 nodes and 731,772 edges, as illustrated in the schematic diagram in Figure 5.

Table 2: Result of the ablation experiment in normalization.

| Method | NER F1 | $\nabla$ | RE F1 | $\nabla$ | ER F1 | $\nabla$ |
|---|---|---|---|---|---|---|
| **Darwin (Normalized)** | 92.96 | - | 88.27 | - | 89.13 | - |
| **ER-N/F** | 92.96 | - | 83.29 | -4.98 | 89.13 | - |
| **ER-NF/A** | 84.01 | -8.85 | 83.07 | -5.20 | 84.54 | -4.59 |
| **ER-ED** | 88.59 | -4.37 | 80.20 | -8.07 | 50.00 | -38.83 |

Finally, to demonstrate the reliability and accuracy of our similarity algorithm, we divided MKG into two knowledge graphs based on the paper's publication year. KG, with earlier years, applied similarity calculation algorithms to predict potential links, while KG, with later years, is used to verify how many sets of "Material-Applications" in these predictions were reported in the following n years. The result is shown in Figure 6. Specifically, in Figure 6 (a), the grey line uses only abstracts published before the year to make predictions, and the percentage of reported predictions in the next few years is displayed. The predictive capacity of the MKG is substantial; therefore, for each prediction, we only use the top 200 "material-application" pairs for validation. Looking at the results, as time progresses, the predicted materials are gradually verified. Using data from 2014, within nine years, 48.5% of the "material-application" predictions have been validated. This outcome demonstrates the effectiveness and reliability of the MKG, and it also lays the foundation for proposing new materials in some fields. The red, blue, and green lines are the average percentage of reported links based on network similarity, Jaccard similarity directly between "Formula" and "Application", and TransE for link prediction, respectively. Network-based similarity has the highest accurate prediction, which

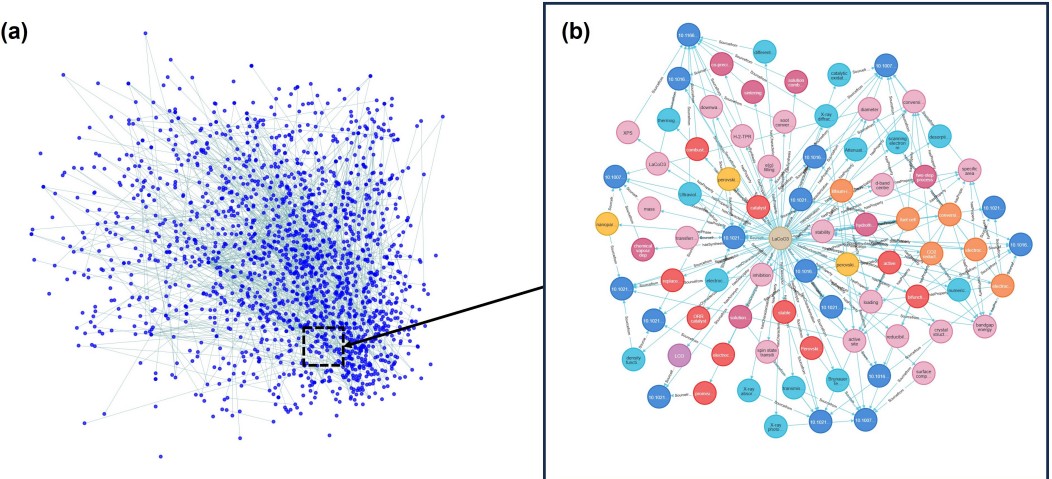

Figure 5: (a) Global schematic diagram of MKG; (b) Local schematic diagram of MKG.

indicates the effectiveness of simulating the thinking of real material development using algorithms. The Figure 6 (b) is an example, visualizing the KG made from data before 2018 and the predicted "Material Application" reported within 5 years using network-based algorithms.

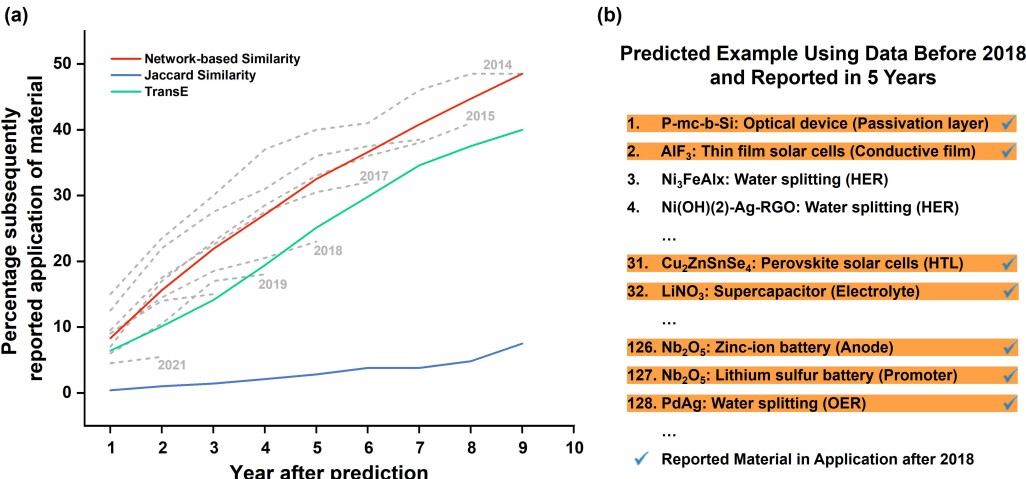

Figure 6: Validation of the graph completion. (a) Percentage of reported prediction after years. (b) Example of predicted material - application using data before 2018.

# 4 Conclusion, Discussion, and Future Work

In this paper, we present a novel natural language processing (NLP) pipeline for Knowledge Graph (KG) construction, designed to efficiently extract triples from unstructured scientific texts. This methodology uniquely enables the fine-tuning of Large Language Models (LLMs) using minimally annotated datasets. The fine-tuned LLMs are then utilized to extract structured information from extensive corpora of unstructured text, bypassing predictive modeling to enhance the authenticity and traceability of the structured data extracted.

Utilizing this approach, we constructed a Material Knowledge Graph (MKG) that encapsulates relations between materials and their associated entities, such as properties and applications, derived from the abstracts of 150,000 peer-reviewed papers. Our analysis not only confirms the effectiveness and credibility of the MKG but also enriches our understanding of the material science domain.

Additionally, our methodology and the resulting KG hold significant potential across various dimensions:

1. Currently, our original text consists of article abstracts, which may omit a substantial amount of experimental methods and data. Fortunately, our method can be directly applied to full-text extraction, allowing us to more accurately capture subtle distinctions within complex scientific literature.

2. The ontology design of MKG has improved the precision of entity labeling in our pipeline and facilitates more accurate categorization of data, including the incorporation of intricate attributes such as synthesis conditions or material properties. This enhancement significantly increases the granularity and practical utility of the KG.

3. The adaptability of our NLP pipeline to a broad range of scientific fields suggests its potential as a foundational template for constructing domain-specific KGs beyond materials science.

4. Integrating our Material Knowledge Graph with existing KGs, such as MatKG, paves the way for the creation of a more interconnected and expansive dataset. This synergy facilitates not only advanced research but also the development of innovative applications in materials science and related domains.

For future work, given the apparent predictive capacity of our method for link prediction, we plan to focus on:

1. Analyzing Historical Patterns: By incorporating additional attributes such as authorship, publication year, and affiliations into our Knowledge Graph, we focus on analyzing historical "social" patterns. For example, investigating whether specific groups or institutions consistently engage in the repurposing of materials. This analysis will allow us to understand how the dynamics of material repurposing are influenced by collaborative networks or institutional trends. We will assess which materials or types of materials are more frequently repurposed and explore the potential social dynamics that drive these trends. This approach aims to provide insights into the lifecycle and innovation patterns of materials based on past activities and collaborations.

2. Identifying Future Materials and Applications: After constructing a complete MKG, in-depth graph neural networks and algorithms can be developed and used to explore potential relations in the graph. By exploring the data within our enriched MKG, we will identify new potential applications for the most promising materials or explore the future trend of materials. This proactive approach seeks to unlock new uses for materials before they become mainstream, providing a competitive edge in materials science. Additionally, by incorporating time dynamics, we aim to enhance the utility of our pipeline, allowing for time-aware graph embedding techniques for graph completion. This will enable us to explore how material properties and applications evolve over time, providing a dynamic perspective on the development and use of materials in various fields.

3. Studying Cluster Formation: We plan to analyze the formation of clusters within the KG, which are based on the links between materials. Understanding these clusters will help reveal how materials are interconnected, which can aid in discovering the connections between different materials that were not apparent from isolated studies.

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

# A Appendix

## A.1 Validation for MKG

To validate MKG, we extract the 500 triples that are randomly extracted except the "DOI" and "Domain" nodes, which are informative information, and checked by the experts in relevant material science. After splitting these triples apart, the annotator can obtain 1000 entities and 500 relations. The report from the annotator is shown in Table 3. The labels "Application", "Structure/Phase", "Synthesis" and "Characterization" exhibit 100% accuracy in both entity and relation. This is due to the rigorous normalization of entities under these categories using our expert dictionary. However, strict normalization can also lead to certain entity loss, which is difficult to avoid. In contrast, the "Descriptor" and "Property", characterized by their vast diversity and broad spectrum, apply a less stringent standardization process, which leads to a certain degree of imprecision in entity and relation. But the results are still satisfactory. "Name" and "Acronym" have also achieved a commendable 100% entity accuracy. However, when it comes to analyzing relations, the ChemDataExtractor encounters certain limitations. A detailed analysis of "Name" and "Acronym" misclassifications reveals that most of these wrong entities originate from "Formula", which is due to the reasonable error of LLM"s binary classification of "Formula" and "Name" and ChemDataExtractor. Fortunately, the impact of this kind of error is not significant and will not bring a significant influence, as fundamentally, the "Name", "Formula" and "Acronym" from same source all represent the same material.

Table 3: Human evaluation metric on randomly selected triples.

| Label | Number in total | Entity disagree | % disagree | Relation disagree | % disagree |
|---|---|---|---|---|---|
| Formula | 257 | 0 | 0 | N/A | N/A |
| Name | 227 | 0 | 0 | 12 | 5.3 |
| Acronym | 26 | 0 | 0 | 3 | 11.5 |
| Descriptor | 127 | 7 | 5.5 | 12 | 9.4 |
| Property | 147 | 12 | 8.2 | 9 | 6.1 |
| Application | 73 | 0 | 0 | 0 | 0 |
| Structure/Phase | 36 | 0 | 0 | 0 | 0 |
| Synthesis | 50 | 0 | 0 | 0 | 0 |
| Characterization | 57 | 0 | 0 | 0 | 0 |

## A.2 prediction result

As Figure 7 illustrates, we display the materials ranked in the top five for 2023 across various domains, alongside their rankings in previous years based on available data up to 2023. It is important to note that these materials-application pairs were not ranked within the top five before 2017, which is why they do not appear in the figure for those earlier years. Stars mark the years these materials were featured in literature for their respective applications. The figure emphasizes the significance and emerging utility of these materials by highlighting the top five in applications such as catalysts, batteries, fuel/solar cells, and other fields from 2017 to 2023, indicating their growing importance and potential to revolutionize technology in these areas. This prediction serves as an inspiration to materials researchers, accelerating progress in material development. For example, consider the material NiFeN HC/NF; it not only excels in the Oxygen Evolution Reaction (OER) but has also demonstrated its capability as a catalyst for the Hydrogen Evolution Reaction (HER). This means that NiFeN HC/NF is a promising bifunctional electrocatalyst for water electrolysis, aligning with our understanding of its potential in dual-function applications.

## A.3 Paramaters for LLM fine tuning

epochs: 10 train batch size (per device): 1 eval batch size (per device): 1 gradient accumulation steps: 2 evaluation strategy: "no" save strategy: "steps" save steps: 500 save total limit: 1 learning rate: 2e-5 weight decay: 0. warmup ratio: 0.03 lr scheduler type: "cosine" logging steps: 1 tf32: False

fsdp: "full shard auto wrap" fsdp transformer layer cls to wrap: 'LlamaDecoderLayer' token max length: 1024

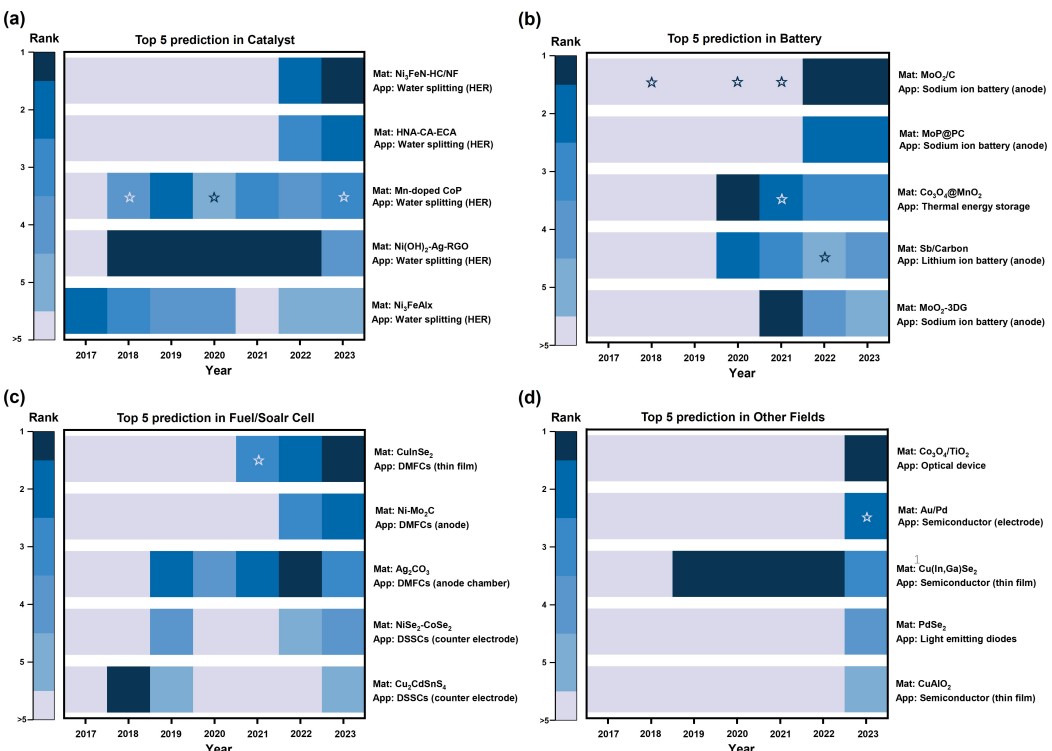

Figure 7: Top five potential materials in (a) Catalyst, (b) Fuel/Solar cell, (c) Battery and (d) Other fields, predicted on all available data (2014 - 2023) and their rank trends in the past years (2014 - related year). The stars represent the material has been reported in the related year.

## A.4 Experiments compute resources

The compute resources were primarily consumed in LLM fine-tuning, which involved using 8 GPUs for 9.02 hours with a 36% utilization rate, 25.98 service units, and 14245 MiB of storage from a quota of 81920 MiB.

## A.5 Data and Code

The MKG is made publicly available at https://doi.org/10.6084/m9.figshare.25997188.v3 in both RDF and CSV format. All the code and datasets for LLM fine-tuning, inference, Entity Resolution and Graph Completion can be found in https://github.com/MasterAI-EAM/Material-Knowledge-Graph.git.

