# OpenReview forum: "Construction and Application of Materials Knowledge Graph in Multidisciplinary Materials Science via Large Language Model"
_NeurIPS.cc/2024/Conference — NeurIPS 2024 poster_

### Official Review · Reviewer_5h8v · 2024-07-09

**Soundness:** 4
**Presentation:** 4
**Contribution:** 4
**Rating:** 7
**Confidence:** 4

**Summary:**

This paper introduces the Materials Knowledge Graph, a pioneering graph database designed for materials science. It leverages advanced NLP methods and LLMs to extract and organize a vast amount of high-quality research into structured triples. It streamlines the discovery process by organizing information into nodes and edges, enhancing data integration and reducing the need for traditional experiments. The MKG also employs algorithms to predict material applications, offering a significant advancement in accelerating materials research.

**Strengths:**

1.	This paper demonstrates a thorough and systematic methodology, including data preparation, model training, entity resolution, and graph construction, ensuring a robust and credible knowledge graph.
2.	The MKG shows a cutting-edge approach to parsing and structuring vast amounts of scientific literature, offering a significant advancement in accelerating materials research.
3.	The application of link prediction algorithms for predicting material applications is a robust method for identifying new potentials in the field.
4.	Experiments shows the effectiveness of the MKG and the employed models, enhancing the credibility and reliability of the results.

**Weaknesses:**

1. As the field of materials science evolves, maintaining the currency and accuracy of the MKG could become increasingly complex, requiring continuous updates and curation.

**Questions:**

1.	How can your methods be adapted for other scientific fields? Are there specific modifications needed?
2.	What strategies do you have for continuously updating and curating the MKG to ensure its relevance and accuracy?

**Limitations:**

The authors have made a commendable effort in addressing the limitations of their work. They acknowledge the dependency on manual annotation for data preparation, which could limit scalability and timeliness.
However, the paper could benefit from a more detailed discussion on maintaining the currency and accuracy of the MKG as the field of materials science evolves. Continuous updates and curation will be crucial, and outlining specific strategies for this would strengthen the paper.

---

> ### Author Rebuttal · Authors · 2024-08-06
>
> We would like to express our gratitude for the thorough review and valuable feedback on our manuscript. Your insights are highly appreciated and have been carefully considered to enhance our work. In this text, we will address each of your comments in detail, clarifying how we have addressed or plan to address the concerns raised.
>
> **Question 1: How can your methods be adapted for other scientific fields? Are there specific modifications needed?**
>
> Response: We appreciate this insightful question. Constructing a knowledge graph requires a well-defined ontology and a practical knowledge extraction method. To adapt our approach to other scientific fields, one primarily needs to design the ontology of the graph. This involves defining the types of nodes and the relations between nodes, as well as annotating a dataset. About 50 data based on our training template should suffice to establish a relatively accurate domain-specific knowledge graph.
> In the ER process, it is advisable to utilise an embedding model particularly suited to your domain. For material science, where properties and applications are deeply influenced by complex material characteristics, additional enhancements should include integrating specialised embedding models tailored to capture these detailed interactions. These models enhance ER by embedding all entities within the materials science context, thus ensuring more accurate and contextually relevant data processing. Notably, while these enhancements are specialised for materials science, they build upon standard foundational models, employing pre-trained frameworks such as word2vec - mat2vec, BERT - MatBERT. These models not only meet the intricate demands of materials science, but also retain the flexibility to be applied broadly across diverse scientific domains. This ensures that our approach, while adaptable to the unique demands of materials science, remains universally applicable. In other words, when there are no suitable domain-specific models, the base models are also applicable.
>
> **Question 2: What strategies do you have for continuously updating and curating the MKG to ensure its relevance and accuracy?**
>
> Response: Response: We thank the reviewer for their comment. We acknowledge the critical importance of keeping the MSKG up-to-date with the latest advancements in the field. To achieve this, we have implemented several strategies for continuous updates. These include the automated monitoring of new publications using predefined keywords and topics, conducted periodically, along with semi-annual manual reviews by domain experts who validate and refine the graph's content. Additionally, we are harnessing advancements in Large Language Models, particularly the development of intelligent agents. These agents actively participate not only in adding new data but also in critically reviewing and correcting existing entries by identifying discrepancies between the updated graph and previous versions. This dual strategy of addition and correction plays a vital role in enhancing the accuracy and relevance of our knowledge graph.
>
> Additionally, to further optimise the pipeline and reduce costs. We also focus on technological innovations that streamline the process of KG construction. Our approach includes automatically constructing an expert dictionary for entity resolution and leveraging clustering algorithms to identify central nodes and pivotal connections within the graph. This method facilitates a more automated and efficient process. Automating these crucial steps significantly reduces reliance on manual curation while ensuring that our knowledge graph remains robust and comprehensive.
>
> **Supplementary explanation:**
>
> We greatly appreciate the reviewer's insights and comments. As mentioned by the reviewer, further enhancing the currency and accuracy of MKG is an essential goal for our future research. With sufficient currency in the knowledge extraction process, our next focus will be on improving entity resolution (ER) by replacing the machine learning model with Large Language Models (LLMs) to further improve the currency of the method. In addition, due to the existence of LLM, the amount of data required for the training set has been significantly reduced. We further use an active learning strategy to create a new training set through each round of knowledge extraction, further reducing labour costs.

---

### Official Review · Reviewer_is27 · 2024-07-09

**Soundness:** 3
**Presentation:** 2
**Contribution:** 3
**Rating:** 7
**Confidence:** 4

**Summary:**

This paper presents an innovative way on leveraging the power of Large Language Models for the construction of a Material Knowledge Graph (MKG) and link prediction. The method includes annotating few scientific articles (abstracts) related to material science which are used for training and finetuning LLMs. After that step and using additional articles and a finetuned LLM, triples are generated. Entity resolution is performed by using different Natural Language Processing (NLP) techniques such as ChemDataExtractor, mat2vec and an expert dictionary. Finally, the MKG is constructed and used for link prediction with the aid of network-based algorithms and graph embeddings. This approach is compared with another technique called MatKG2 and experiments were conducted for finding the LLM that provides the best results and for evaluating the link prediction of the MKG.

**Strengths:**

•	This is an innovative work for the fast and automatic construction of a Material Knowledge Graph with minimal annotation that could have a broad impact in the advancements of material science.

•	It reuses effectively new technologies such as Large Language Models and other NLP techniques.

•	It does not require too many annotated documents or other manual tasks.

**Weaknesses:**

•	More experiments would have supported better this work. The result of this method could have been compared with some baseline experiments of simply using LLMs for the triples generation. Moreover, since this method is compared with MatKG2, it would have been useful to compare the results of precision, recall and F1 score of MatKG2 with the results of MKG.

•	While the work seems quite interesting it not well written and several typos and syntactic errors have been found which are detailed below. Proof-reading would have been beneficial before submission.

- Line 64: “ A user-friendly databases.. “ -> “User-friendly databases..”

- Line 80: Acronyms NER and RE are used but they are introduced later in the text (line 85)

- Line 89: “… through query the MKG.” -> “… through querying the MKG.”

- Line 93: “the elaborate workflow” -> “the elaborated workflow”

- Line 98: “NERRE” -> I believe this refers to NER and RE

-	Figure 1 (b): ChemDataExactor -> ChemDataExtractor

- Figure 4: FMKG -> the material knowledge graph has been referred in the whole paper MKG. The acronym FMKG is introduced only in this figure.

- Line 233: “indicate”->”indicates", “achieve”->”achieves”, “contribute”->contributes”.

- Line 240: “ChemDataExactor” -> ”ChemDataExtractor”

- Line 252: “The result shows in.. “ -> “The result is shown in…”

• Figure 3 and network-based link prediction is not well explained in the article.

• The code and Knowledge Graph is not available for further evaluation. The authors state that they will only be available if the paper gets accepted.

**Questions:**

Limitations of this work are not detailed in the paper.
- What would you consider limitations of this work?
- How would you ensure that MKG will be up-to-date with the state of the art in material science?

**Limitations:**

There are some concerns about the scalability and maintainability of MKG which are not addressed in the paper.

---

> ### Author Rebuttal · Authors · 2024-08-06
>
> We would like to express our gratitude for the thorough review and valuable feedback on our manuscript. We will answer all your questions and concerns, clarifying how we have addressed the concerns raised.
>
> **Question 1: What would you consider limitations of this work?**
>
> Response: We would like to thank the reviewer for their comment. The limitation of the work is that the entities and relations in the Entity Resolution task are missing. As noted in the 'Appendix,' we prioritise accuracy in our knowledge graph, and thus, we implement rigorous normalisation processes in ER. This approach may inadvertently lead to the wrongful exclusion of some correct entities. To mitigate this, we are planning for a multi-tier knowledge graph approach. In this approach, entity processing is staged across multiple levels. Initially, entities are processed with broader, less restrictive criteria. As they progress through the system, increasingly stringent criteria are applied at each level to refine and purify the entity data. Additionally, entities that are initially filtered out undergo a secondary review process using LLMs, which helps determine whether to reintegrate them or permanently exclude them. This staged, iterative process aims to balance accuracy with coverage, minimising wrongful exclusions while maintaining the integrity of our knowledge graph.
>
> **Question 2: How would you ensure that MKG will be up-to-date with the state of the art in material science?**
>
> Response: We would like to thank the reviewer for their comment. We agree that this is a crucial aspect of our future work. To ensure that MSKG remains current with the state of the art in materials science, we have implemented several strategies for continuous updates. These include automated monitoring of new publications using predefined keywords and topics, conducted periodically, and semi-annual manual reviews by domain experts to validate and refine the graph content. Additionally, we are incorporating advancements in LLMs, particularly the development of Agents. These agents not only add new data but also critically review and correct existing information by identifying discrepancies between the updated graph and previous graph. This dual approach of addition and correction significantly enhances the accuracy and relevance of our knowledge graph.
>
> **Weakness: More experiments would have supported better this work...**
>
> Response: We have tried to add several naive LLMs into baseline experiments, However, with fine-tuning, LLM performs better, but also cannot perform NER and RE stably according to the expected format. That's why we gave up adding simple LLM. To further elaborate on the reasons, we cited a work that can prove this viewpoint and have made modifications in the article: “We evaluated the performance of each LLM on each task…” → “We evaluated the performance of each LLM across various tasks. Xie et al. *(Tong Xie, Patterns)* have demonstrated by comparing GPT-3.5 with the fine-tuned LLaMA that the latter significantly outperforms the naive models. Even under more lenient manual evaluation conditions for GPT-3.5, a noticeable performance gap persists between it and the fine-tuned LLaMA. This evidence aligns with our findings.” Therefore, we exclude the naive LLMs from the evaluation baseline.
> The purpose of comparing with MatKG2 is to highlight the traceability of knowledge in MKG, reflecting the optimisation of the process. Given that the relation in MatKG2 is predicted, we believe a quantitative comparison of the construction methods between MatKG2 and MKG would be unfair. Moreover, the data and code of MatKG2 is not available online. However, we fully acknowledge the reviewer's suggestion. Therefore, we have included the evaluation of MatBERT for knowledge extraction which is core of the MatKG construction, and relevant discussion in the baseline. We think this additional experiment can not only provide a comparison between the MKG and MatKG series but also provide a comparison between the LLM and non-LLM pipelines. The results of this expanded baseline comparison will be detailed in our next revision and we have put the result in global response for your reference.
>
> *Xie, Tong, et al. "Creation of a structured solar cell material dataset and performance prediction using large language models." Patterns 5.5 (2024).*
>
> **Weakness: Typo and code.**
>
> Response: We thank the reviewer for pointing out the typo. We have carefully checked the paper again to ensure that all the typos and syntactic errors have been modified. All the code and data are open-source, and the link will be added once this work is accepted.
>
> **Weakness: Extension for the Fig 3 and network-based link predication.**
>
> Response: We would like to thank the reviewer's comment. We have replaced the original caption for Fig 3 with a new caption: "**Fig 3:** The process of network-based graph completion, illustrating how nodes are categorised into Materials, Properties, and Applications. The diagram on the left shows that both old and new Materials share similar Properties as well as similar Applications. As depicted in the centre diagram, this shared attribute implies a degree of similarity between the old and new Materials - similar characteristics and applications. Consequently, as shown in the right diagram, old Materials can potentially be utilised in new Applications."
>
> **Limitation: There are some concerns about the scalability and maintainability of MKG which are not addressed in the paper.**
>
> Response: Our current work aims to find an outstanding pipeline to construct the KG, so we have relatively simplified some content, such as focusing on the abstract of papers. In future work, the scalability and maintainability of the knowledge graph are important tasks, including but not limited to broadening the ontology, including quantitative data, integrating with existing materials science databases, and dynamically updating the knowledge graph.

---

> > ### Comment · Reviewer_is27 · 2024-08-11
> >
> > Thank you for the clarifications and additional comparison results.
> > My concerns have been addressed and I am raising my score.

---

> > > ### Author Response · Authors · 2024-08-13
> > >
> > > We appreciate the reviewer’s constructive comments and welcome further input to enhance our paper’s quality in the time ahead.

---

### Official Review · Reviewer_xvtf · 2024-07-13

**Soundness:** 3
**Presentation:** 3
**Contribution:** 3
**Rating:** 7
**Confidence:** 5

**Summary:**

The study presents an innovative pipeline for Knowledge Graph (KG) construction, specifically designed for efficient extraction of triples from unstructured scientific texts. The methodology enables fine-tuning of Large Language Models (LLMs) with limited annotated datasets, which is then utilized to extract structured information from extensive corpora of unstructured text. The authors have constructed a Material Knowledge Graph (MKG) that captures relationships between materials and their associated entities, such as properties and applications, derived from abstracts of 150,000 peer-reviewed papers.

**Strengths:**

Originality: The paper introduces a novel pipeline for KG construction that departs from predictive modeling, enhancing the authenticity and traceability of extracted structured data.

Quality: The authors demonstrate the effectiveness and credibility of the MKG through ablation experiments and similarity analyses based on node similarity and graph embedding. The results indicate the substantial predictive capacity of the MKG, with 48.5% of 'material-application' predictions validated within nine years, which is impressive.

Clarity: The paper provides a clear and comprehensive explanation of the methodology, including the fine-tuning of LLMs, extraction of structured information, and construction of the MKG. Detailed results and analyses support the authors' claims.

Significance: The MKG has significant potential in extending the depth of structured information extraction, improving entity labeling precision, and adapting the pipeline to other scientific fields.

**Weaknesses:**

Some figure captions, like "Fig 1(a)" and "Fig 3," are unclear. Captions would benefit from additional context.

Certain acronyms, such as "ER-NF" on line 239 and "NER” / “RE" on line 80, appear before being defined.

Typo on line 140: "task"s->task’s."

**Questions:**

Does the inclusion of DOIs affect the resource consumption such as memory or performance in any noticeable way?

It is common practice to begin with abstracts. Do the authors intend to extend their work to encompass full texts in the future?

**Limitations:**

The authors did addressed the limitations, and mentioned that strict normalization and entity resolution process can loss some correct entities.

---

> ### Author Rebuttal · Authors · 2024-08-06
>
> We would like to express our gratitude for the thorough review and valuable feedback on our manuscript. In this text, we will address each of your comments in detail, clarifying how we have addressed or plan to address the concerns raised.
>
> **Question 1: Does the inclusion of DOIs affect the resource consumption such as memory or performance in any noticeable way?**
>
> Response: We would like to thank the reviewer's comment. The inclusion of DOI nodes indeed increases the total number of nodes within the knowledge graph, which in turn marginally elevates memory usage. However, the increase is manageable and does not stress modern hardware. We have performed stress tests to confirm that the additional memory requirements are well within the capabilities of modern hardware systems (Actually, the entire MKG has only 70 MB when converted into RDF format).
> Regarding performance, the knowledge graph is a relational structure database that enables selective interactions between nodes. For example, during operations such as calculating the shortest paths, nodes connected by the 'Sourcefrom' relation can be selectively excluded from computations. This strategic exclusion eliminates any potential adverse impacts on the graph's performance.
> Additionally, we have other strategies to mitigate the influence of the increase in nodes, such as storing DOI nodes and other nodes separately. The subgraph of DOI is only queried when users specifically require information related to these DOIs, ensuring efficient data management and system performance.
>
> **Question 2: It is common practice to begin with abstracts. Do the authors intend to extend their work to encompass full texts in the future?**
>
> Response: We would like to thank the reviewer for their comment. We intend to extend our work to encompass full texts, which is an important part of our future work. In addition to broadening the scope of our analysis, we plan to refine our ontology design to include more comprehensive information. This will involve selectively extracting and standardising quantitative data from scientific articles. We also aim to implement a hierarchical approach to manage the complexity and enhance the efficiency of information extraction from full texts. For instance, we could initially categorise the text into sections such as 'Introduction,' 'Methods,' 'Results,' and 'Discussion.' This segmentation allows us to apply specific extraction techniques tailored to the information typically found in each section, streamlining the process and improving accuracy.
>
> **Weakness:**
>
> We thank the reviewer for highlighting the issues with our figure captions and typo errors. We have made the necessary corrections throughout the manuscript. For instance, the original caption for Fig 3 is replaced by: "**Figure 3:** The process of network-based graph completion, illustrating how nodes are categorised into Materials, Properties, and Applications. The diagram on the left shows that old and new Materials share similar Properties and similar Applications. As depicted in the centre diagram, this shared attribute implies a degree of similarity between the old and new Materials - similar characteristics and applications. Consequently, as shown in the right diagram, old Materials can potentially be utilised in new Applications."

---

> > ### Comment · Reviewer_xvtf · 2024-08-12
> >
> > The explanations regarding the impact of DOIs on resource consumption and performance are reassuring. I appreciate the corrections you made to figure captions and typo errors. Overall, I am confident that your manuscript is technically solid and has high impact on the field, and I recommend its acceptance for publication.

---

> > > ### Author Response · Authors · 2024-08-13
> > >
> > > We appreciate the reviewer’s constructive comments and welcome further input to enhance our paper’s quality in the time ahead.

---

### Official Review · Reviewer_jF92 · 2024-07-15

**Soundness:** 3
**Presentation:** 3
**Contribution:** 3
**Rating:** 4
**Confidence:** 4

**Summary:**

The paper on constructing and applying a materials knowledge graph (MKG) in multidisciplinary materials science via a large language model (LLM) is valuable and well-written. However, it primarily focuses on application rather than strong technical contributions, with issues in experimental design, lack of non-LLM baselines for comparison, and insufficient comparison with sophisticated knowledge graph completion methods. It may be a good dataset track paper but not the main track.

**Strengths:**

- The studied problem is of great value in the real world
- The paper is well-written and easy to follow
- The produced MKG could be very helpful for the computational and experimental material science

**Weaknesses:**

- W1: This paper is towards applications of LLMs in an engineering flavor instead of having strong technical contributions. The process includes prompt engineering, basic model fine-tuning, and human-in-the-loop entity resolution. And this paper looks like an extended application of Darwin, instead of an independent research work.
- W2: The experimental design is flawed. In particular, why the normalization is only applied to Darwin? Is it possible that other base models + normalization can perform better? And is the normalization only working well for Darwin?
- W3: Non-LLM baselines for NER and RE should be included for comparison.
- W4: The modified Jaccard similarity method is claimed as a specified KG completion algorithm for material science. Therefore, the experiments should include comparisons with more sophisticated KG completion methods, instead of only comparing with TransE, which is outdated.
- W5: Minor issues include but are not limited to: Typos in Figure 4, what is FMKG?

**Questions:**

Please refer to W2 and W5

**Limitations:**

Yes.

---

> ### Author Rebuttal · Authors · 2024-08-06
>
> We would like to express our gratitude for the thorough review and valuable feedback on our manuscript. Your insights are highly appreciated and have been carefully considered to enhance our work. We also want to emphasise some points in the paper to answer your question.
>
> **W1 Response:**
>
> We would like to thank the reviewer's comment. As reviewers xvtf and is27 have noted, this paper emphasises a novel process for constructing a materials science knowledge graph. The novelty of our work lies in integrating state-of-the-art LLM to perform NERRE simultaneously and for continuous and dynamic updating of the knowledge graph, allowing for real-time integration of new research findings directly into the MKG. The challenges include the potential for LLM-generated hallucinations and biases during the knowledge extraction process, which necessitates sophisticated normalisation and entity resolution processes to maintain the integrity and credibility of the knowledge graph. Additionally, it is crucial to emphasise that this is the first knowledge graph in the materials science domain that achieves multi-level connections for complex material science knowledge, representing a significant advancement over previous models that were limited to merely binary connections. For instance, instead of linking an alloy’s composition directly to its mechanical properties, MKG includes intermediate nodes such as processing techniques and microstructural features. This allows for a deeper exploration of the complex interdependencies that affect an alloy's properties, offering an unprecedented level of detail in modelling material interactions.
>
> Solving STEM problems through LLMs may seem like an engineering flavour, but it is indeed a research direction encouraged by *NeurIPS*, such as the work of Hu et al. [1] and Lu et al. [2]. Compared with the biomedical field, the rarity of KG applications in material science is primarily due to the absence of effective knowledge graphs that can comprehensively capture the intricate data and complex relations inherent in these fields. Different from existing material science knowledge graph construction methods, such as the MatKG series, we utilise LLM for knowledge extraction, ensuring authenticity and traceability of the information. In other words, this work is not an extension of Darwin, although Darwin's performance is the best among several open-source models we compared.
>
> [1] Hu, Xiuyuan, et al. "De novo drug design using reinforcement learning with multiple gpt agents." Advances in Neural Information Processing Systems 36 (2024).
>
> [2] Lu, Pan, et al. "Learn to explain: Multimodal reasoning via thought chains for science question answering." Advances in Neural Information Processing Systems 35 (2022): 2507-2521.
>
> **W2 Response:**
>
> In this study, the application of LLMs is primarily focused on the knowledge extraction part, where we finetuned LLMs on a small set of annotated data to effectively perform entity and relation extraction. The following normalisation process involves cleaning and standardising NERRE result. Therefore, once we identified the LLM with the best performance in knowledge extraction (i.e., Darwin for this task), we concentrated on normalising its output. Theoretically, normalisation is applicable to any LLM that can successfully perform knowledge extraction. However, in our study, we opted for the best-performing model for in-depth analysis based on considerations of efficiency and clarity.
>
> We have also revised the caption to make our ideas more straightforward and avoid misunderstandings: “*Table 1:* Comparative results of NER, RE and ER across different models using fine-tuned LLMs and non-LLMs. The Darwin model, which demonstrated the highest overall performance, was selected to showcase the effects of subsequent normalisation.”
>
> **W3 Response:**
>
> Initially, we focused on leveraging LLM due to their advanced capabilities in handling the complex semantics of scientific texts, which is critical for both NER and RE. Besides, the low labour cost required for fine-tuning is also an indispensable factor in building an automated pipeline. In contrast, identifying the relations by non-LLMs requires a large amount of accurately annotated data for training, which could not be efficient when used for a dynamic knowledge graph that includes state-of-the-art domain knowledge. The absence of non-LLM models in our initial baseline was due to this strategic focus rather than an oversight. In response to the reviewer's insightful feedback, we have incorporated MatBERT and MatKG's construction methods. The results of this expanded baseline comparison will be detailed in our next revision and we have put the result in global response for your reference.
>
> **W4 Response:**
>
> We would like to clarify that the primary focus of our research is leveraging LLMs for KG construction, which addresses issues during the construction process rather than on graph completion itself. The purpose of employing graph completion techniques is to demonstrate the quality, credibility, and effectiveness of the KG we constructed. In other words, using both Jaccard and TransE was not intended to compare their effectiveness; instead, our goal was to explore different methods to enhance the credibility of these steps. Choosing these well-known methods ensures that the study remains approachable and comparable. We opted to modify the basic Jaccard algorithm due to the suboptimal performance of the standard similarity algorithm, as depicted in Fig 6. This modification reflects a common practice in materials science research, where researchers often select trending materials from fields aligned with their research areas as potential candidates for current applications. Our modifications, simulating this process, were minor yet crucial to better suit the needs of the MKG.
>
> **W5 Response:**
>
> We apologise for this typo, and thanks for pointing out this problem. it was revised to “MKG”.

---

> > ### Comment · Reviewer_jF92 · 2024-08-13
> >
> > Thanks for the rebuttal, it solves some of my concerns.
> >
> > Still,
> >
> > 1. It would be great if the authors could include non-LLM baselines, even on subsets of the data is valuable. It is hard to determine the effectiveness without comparisons to non-LLM methods.
> >
> > 2. It is important to show the proposed normalization can be (easily) applied to and can work well with other models except for Darwin because foundational models are rapidly getting better.
> >
> > I acknowledge the potential benefits that this paper can bring to the community, while the above-mentioned drawbacks have to be improved.
> >
> > Therefore I will keep my evaluation of the paper, according to its current state.

---

> > > ### Author Response · Authors · 2024-08-14
> > >
> > > Thank you for your continued engagement and constructive feedback. We appreciate your insights and have addressed your concerns as follows:
> > >
> > > **1. Inclusion of Non-LLM Baselines:** As per your suggestion, we have already included results from MatBERT, which is a SOTA model for material science tasks, in our experiments for KG construction. Please find the results in the global author rebuttal.
> > >
> > > **2. Normalization:** According to your suggestion, we have broadened our experimental scope to apply the normalization for LLaMA2. This additional experiment demonstrates the simplicity of applying normalization and its effectiveness on other models. If you deem it necessary, we will include the normalization results for each model in the final version of our manuscript and add relevant discussion. As we have mentioned, implementing this process is not challenging.
> > >
> > > | Model | Task | Precision | Recall | F1 score |
> > > |-------|------|-----------|--------|----------|
> > > |       | NER  | 0.9331    | 0.9145 | 0.9237   |
> > > | LLaMA2 (Normalization) | RE   | 0.8517    | 0.8893 | 0.8701   |
> > > |       | ER   | 0.9164    | 0.8902 | 0.9031   |
> > >
> > > We hope these additions and clarifications address your concerns adequately. We believe that these improvements significantly enhance the contribution of our work to the community and appreciate your suggestions that led to these refinements.

---

### Author Rebuttal · Authors · 2024-08-06

We would like to express our sincere appreciation to all the reviewers for your valuable feedback on our work, and we have responded to all your questions (in the corresponding rebuttal sections). We also add some supplementary experimental results in the *pdf* file, mainly to reproduce the KG construction method used by MatKG. Since this method is centered around MatBERT, we think this additional experiment can not only providea comparison between the MKG and MatKG series but also provide a comparison between the LLM and non-LLM pipelines.

---

### Decision · Program_Chairs · 2024-09-25

**Decision:**

Accept (poster)

**Comment:**

This paper presents an approach for using LLMs to construct a Material Knowledge Graph (MKG). The review point out multiple strengths of the paper, including real-world value of the studied problem, the innovative KG construction approach, the systematic methodology used and the use of link prediction in this context. Weaknesses revolve around the paper potentially being light on technical contributions. Overall, the strengths outweigh the weaknesses and I recommend acceptance of the paper.